# The Miocene subsidence pattern of the NW Zagros foreland basin reflects the southeastward propagating tear of the Neotethys slab

Renas I. Koshnaw[1, a], Jonas Kley[1], Fritz Schlunegger[2]

[1]Department of Structural Geology and Geothermics, Geoscience Center, University of Goettingen, Goldschmidtstrasse 3, 37077 Göttingen, Germany
[2]Institute of Geological Sciences, University of Bern, 3012 Bern, Switzerland

[a]*Correspondence to*: Renas I. Koshnaw (renas.koshnaw@geo.uni-goettingen.de)

**Abstract.** Tectonic processes resulting from solid earth dynamics control uplift and generate sediment accommodation space via subsidence. Unraveling the mechanism of basin subsidence elucidates the link between deep earth and surface processes. The NW Zagros fold-thrust belt resulted from the Cenozoic convergence and subsequent collision between the Arabian and Eurasian plates. The associated Neogene foreland basin includes ~3-4 km of synorogenic, mostly nonmarine clastic sediments, suggesting a strongly subsiding basin inconsistent with the adjacent moderate topographic load. To explain such discrepancy, we assessed the magnitude of the basin's subsidence with respect to the effect of surface load and dynamic topography. The lower Miocene isopach map of the Fatha Formation displays a longitudinal depocenter aligned with the orogenic trend. In contrast, the middle-late Miocene maps of the Injana and Mukdadiya formations illustrate a focused depocenter in the southeastern region of the basin. This rapid basin subsidence in the southeast during the middle-late Miocene was coeval with the Afar plume northward flow beyond the Arabia-Eurasia suture zone in the northwestern segment of the Zagros belt. Based on isopach maps, subsidence curves, and reconstructions of flexural profiles, supported by Bouguer anomaly data and maps of dynamic topography and seismic tomography, we argue for a two-stage basin evolution. The Zagros foreland basin subsided due to the combined loads of the surface topography and the subducting slab during the early Miocene and was affected by dynamic topography due to the Neotethys horizontal slab tear propagation during the middle-late Miocene. This tear propagation was associated with a northward mantle flow above the detached slab segment in the NW and a focused pull on the attached portion of the slab in the SE.

## 1 Introduction

The formation of fold-thrust belts and the evolution of the adjoining basins may take tens of millions of years and involve large-scale processes such as deformation, magmatism, metamorphism, uplift, and subsidence that are collectively coupled with lithosphere dynamics and mantle flow (Royden, 1993; Allen and Allen, 2013). At convergent plate boundaries, accommodation space is created by the flexural bending of the foreland plate due to surface and slab loading (Schlunegger and

Kissling, 2022). The history of such a basin could be influenced by mantle flow and the evolution of dynamic topography (Dávila and Lithgow-Bertelloni, 2013; Jolivet et al., 2015; Heller and Liu, 2016). During orogenesis, continents converge through two main mechanisms to form a mountain belt and create a foreland basin: slab-pull orogeny, driven mainly by the weight of the downgoing slab, and mantle orogeny, driven by shear traction along the bottom of the slab as asthenospheric

mantle flows (Conrad and Lithgow-Bertelloni, 2004; Schlunegger and Kissling, 2015; Royden and Faccenna, 2018; Faccenna et al., 2021). In convergent plate boundaries, particularly in continental collision zones, a slab breakoff may occur. The slab breakoff process affects the orogeny, the development of topography and surface mass flux, and the flow of mantle materials at depth (Sinclair, 1997; Kissling and Schlunegger, 2018; Vanderhaeghe, 2012). Slab breakoff is considered to occur when the buoyant continental lithosphere starts to enter the subduction channel. This results in tensional forces between the subducted

oceanic and continental parts of the plate, with the consequence that the denser subducting oceanic lithosphere detaches from the lighter continental counterpart, and sinks, thereby triggering magmatism, uplift, and the exhumation of metamorphic rocks (Davies and von Blanckenburg, 1995). The process of separation is thought to involve a sequence of processes, including slab necking, tearing and detachment, and eventually slab breakoff (complete detachment of slab) (Kundu and Santosh, 2011). Even though these processes take place at depths of hundreds of kilometers and have not been fully understood (e.g., Niu,

2017; Garzanti et al., 2018), potential surficial consequences through time and space could be constrained by geologic records and numerical models (Sinclair, 1997; van Hunen and Allen, 2011; Duretz and Gerya, 2013; Jolivet et al., 2015; Garefalakis and Schlunegger, 2018; Boutoux et al., 2021). Furthermore, global tomography and the results of numerical models highlight that these processes occur in 3D, and envisioning them in 2D may not provide enough insight (Hafkenscheid et al., 2006; van Hunen and Allen, 2011; Balázs et al., 2021). As slab breakoff occurs along the continental suture zone, essential changes in

the force balance are expected to occur, with the consequence of measurable geologic processes on the surface such as uplift, subsidence, and an increase in the mantle-influenced magmatism (Wortel and Spakman, 2000).

Unlike along the subduction boundary of the Americas, a breakoff of the downgoing Neotethys oceanic slab has been suggested to occur along the collisional boundary of Eurasia beneath the Alps, the Zagros, and the Himalaya. Among these belts, the

Zagros orogen is the youngest and the least deformed belt and has a relatively complete and well–preserved rock record (Hatzfeld and Molnar, 2010). Because the Arabia–Eurasia continental collision commenced in the area of the NW Zagros fold-thrust belt (McQuarrie and Hinsbergen, 2013), this segment of the orogenic belt and the related foreland basin are particularly interesting for inferring the controls on the construction of the orogen and the adjacent foreland basin (Fig. 1). Furthermore, the documented along–strike variations in geological properties near the surface (e.g., activation timing of the Main Zagros

fault (MZF), age of exposed rocks, amount of foreland basin accommodation) and at depth (e.g., lithospheric thickness, age and chemistry of the late Miocene magmatism, and low versus high seismic velocity in the upper mantle) render this orogen an ideal laboratory to analyze the geological record of a time-transgressive orogenesis.

This research in the NW Zagros belt in the Kurdistan region of Iraq aims to constrain the mechanisms by which the Zagros

basin evolved in relation to geodynamic processes such as slab breakoff and mantle dynamics. For this purpose, we generated

new isopach maps and subsidence curves, and evaluated the flexural effect of the Neogene Zagros topography. We synthesized

regional Bouguer gravity anomaly and dynamic topography maps as well as teleseismic tomographic data and the magmatic

record in the Middle East (Amaru, 2007; Hall and Spakman, 2015; Ball et al., 2021) to guide interpretations and analyze the

occurrence of possible surface signals attributable to slab breakoff beneath the orogen.


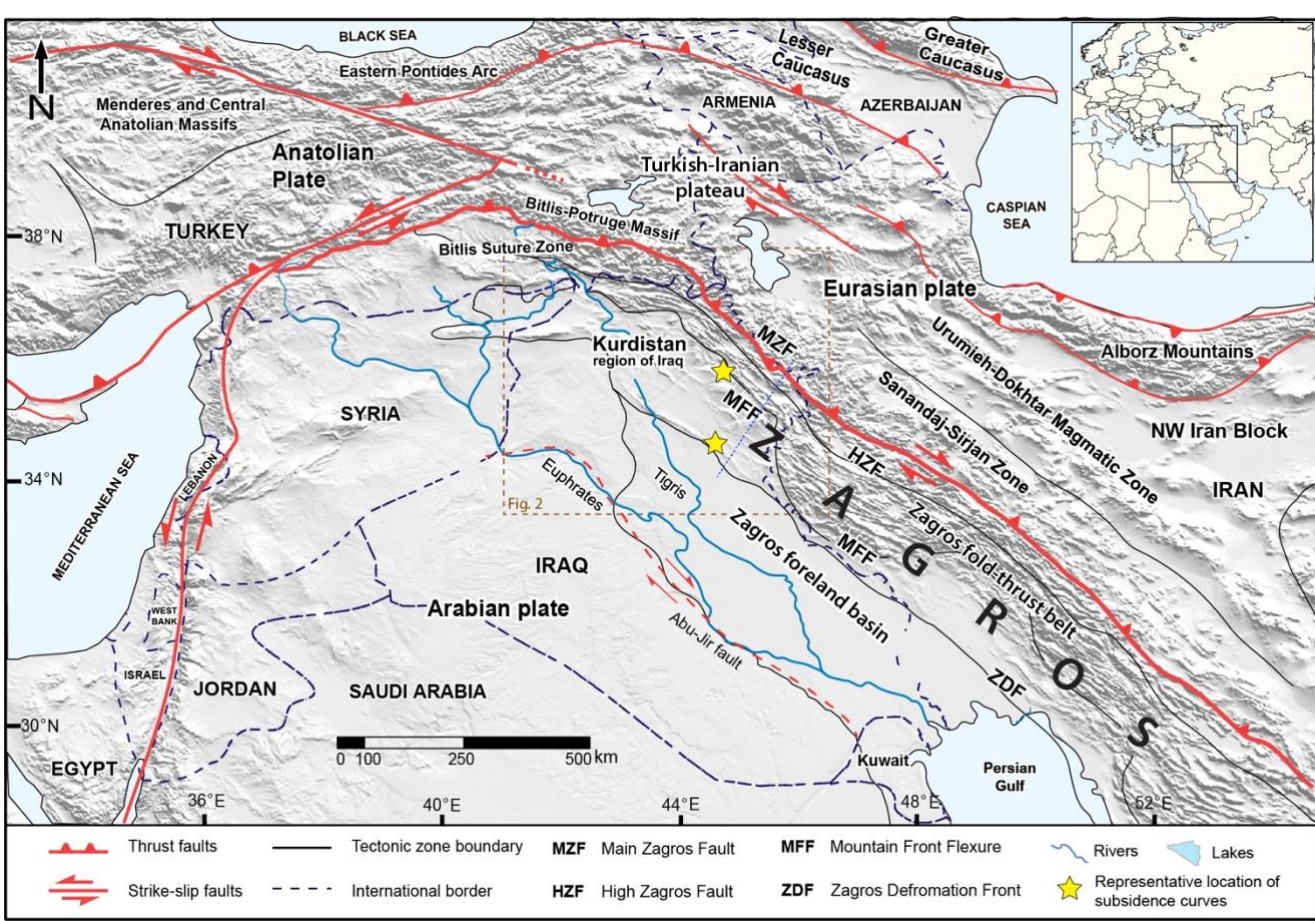

**Figure 1: Tectonic map of the Middle East showing the main structural features, tectonic plates, and the study area in the northwestern segment of the Zagros fold-thrust belt and foreland basin. The dashed brown box represents the study area. The shaded relief map was generated from the ASTER Global Digital Elevation Map (2011) data set (asterweb.jpl.nasa.gov/gdem.asp).**

**2 Tectonostratigraphic context**

After the complete subduction of the Neotethys oceanic slab beneath the Eurasian continental plate, the continental part of the

Arabian plate started to enter the subduction zone and thus collided with the Eurasian plate. Various ages for the Arabia-

Eurasia collision have been suggested, spanning from the late Cretaceous to the Pliocene (Dewey et al., 1973, 1986; Berberian and King, 1981; Dercourt et al., 1986; Hempton 1987; Alavi 1994; Agard et al. 2005, 2011; Fakhari et al. 2008; Ballato et al. 2011; Khadivi et al. 2012; McQuarrie and Hinsbergen 2013; Saura et al. 2015; Gholami Zadeh et al., 2017; Pirouz et al., 2017; Darin and Umhoefer, 2022; Sun et al., 2023). However, as continental collision is a prolonged process, the onset and culmination of the process need to be distinguished. Additionally, the collisional age must agree with well-constrained global paleotectonic models and geologic records. Continental collision is defined by the total subduction of oceanic crust between two continental crusts (Dewey and Horsfield, 1970). In the NW Zagros belt, considering the Arabia-Eurasia collisional age older than the Oligocene is less likely due to (1) the necessity for the occurrence of an unrealistically long post-collisional subduction of the Arabian continental crust beneath Eurasia (e.g., McQuarrie and Hinsbergen, 2013) and (2) the pervasive occurrence of oceanic crust subduction-related magmatism during the Paleocene and the Eocene (e.g., Chiu et al., 2013). Considering that the Arabia-Eurasia collisional age was younger than the Oligocene does not account for lines of evidence from provenance, geochronology, and thermochronology studies, and the timing of crustal deformation as well as regional tectonostratigraphic observations (Allen and Armstrong, 2008; Koshnaw et al., 2019, 2021; Cai et al., 2021; Song et al., 2023).

After the continental collision during Oligocene, the foreland basin situated on the Arabian plate shifted from being underfilled with primarily deeper to shallow marine deposits (ophiolite obduction-related proto-Zagros) to filled and overfilled with mostly nonmarine clastic deposits (collision-related Neogene Zagros) (Fg. 2). Both suites are separated by an Oligocene unconformity, particularly in the hinterland. This is because, before the Neogene, the proto-Zagros fold-thrust belt was already in place as highlands in the northeastern frontiers of Arabia. Along the Arabia–Eurasia suture zone, clastic and carbonate rocks of the Red Beds Series (RBS) were deposited in an intermontane basin on the Arabian plate, situated immediately below the Main Zagros fault (MZF). The RBS strata show an angular unconformity with the older folded strata from the highlands of the proto-Zagros. The RBS consists of the nonmarine clastic Suwais Group, unconformably overlain by the shallow marine carbonate rocks of the Govanda Formation and the nonmarine clastic Merga Group on top. The clastic units of the Merga and Suwais Groups contain provenance signatures from Eurasia and the beds from the lower contact have a maximum depositional age of ~26 Ma (Koshnaw et al., 2019). These units of the RBS were overthrusted by the allochthonous thrust sheets of the Walash–Naopurdan–Kamyaran (WNK) and ophiolitic terranes along the Arabia-Eurasia suture zone.

Away from the suture zone, the present-day NW Zagros Neogene foreland basin includes a ~3-4 km thick succession that consists of the mixed clastic–carbonate–evaporite beds of the Fatha Formation and the clastics of the Injana, Mukdadiya, and Bai–Hasan formations (Fig. 2). These formations constitute classical foreland basin components with an upward coarsening and thickening succession (DeCelles, 2012). Data on detritus provenance and stratigraphic correlations in the NW Zagros belt indicate that the post–Oligocene foreland basin was a contiguous basin up to the latest Miocene. Afterwards, the basin was structurally partitioned due to the out-of-sequence growth of the mountain front flexure (MFF) (Koshnaw et al., 2020a).

During the early Miocene, the Fatha Formation was deposited in a lagoonal environment on the Arabian plate, fringing the NW-SE-trending Zagros fold-thrust belt. This formation is primarily evaporitic in the depocenter, consisting of alternating beds of marls, limestones, and gypsum, whereas marls, limestones, mudstones, and fine- to medium-grained sandstones predominating the basin periphery (Shawkat and Tucker, 1978). In the NW Zagros belt, the thickness of the Fatha Formation increases from the margin (c. 300 m) to the center (c. 600 m) of the basin (Dunnington, 1958; Koshnaw et al., 2020b). Provenance evidence based on detrital zircon U-Pb ages shows that the sandstone component of the Fatha Formation was delivered mainly from the Paleozoic rocks in the north and Paleogene rocks in the northeast (Koshnaw et al., 2020b). Being an incompetent layer in the stratigraphic column, the Fatha Formation became a décollement layer, detaching the post-Fatha units from the underlying stratigraphic units and leading to the genesis of shallow structures in the NW Zagros belt.

In the middle-late Miocene, the clastic Injana Formation was deposited, marking a fundamental change in the depositional environment from marine to nonmarine. During this time, clastic sediments were delivered into the basin by an orogen-parallel fluvial system, transporting fine-grained sediments through meandering and low-sinuosity channel belts (Koshnaw et al., 2020b). The thickness varies significantly from the NW to the SE, increasing from ~300 m to ~1600 m. The detrital zircon U-Pb age signatures and apatite (U-Th)/He data imply that the deposition of the Injana Formation occurred contemporaneously with the uplift of the northern and northwestern terranes beyond the Arabia–Eurasia suture zone, and concomitant with the reactivation of the MZF along the suture zone with a right-lateral strike-slip component (Koshnaw et al., 2020a).

The Mukdadiya Formation was deposited throughout the latest Miocene, with sediments mainly derived from northeastern terranes and transported by a straight, high-energy transverse fluvial system. The thickness of the Mukdadiya Formation varies from ~300 to ~1,000 m. This shift in the fluvial style between the Injana and Mukdadiya formations occurred simultaneously with the advance of the deformation front into the foreland as documented by growth strata. By the Pliocene, the basin was structurally compartmentalized, with basement-involved thrusting resulting in the formation of the mountain front flexure (MFF) (Fig. 2). During this time, the Bai-Hasan Formation was deposited, consisting mainly of conglomeratic beds at proximal sites and more mudstone layers in the distal parts. In proximal positions, the sediments of this formation were likely deposited by alluvial sheet floods during ephemeral runoff, whereas in the distal parts, discharge occurred in alluvial channels that were bordered by floodplains.

## 3 Methods

Isopach maps were constructed for the Fatha, Injana, and Mukdadiya formations, and we also created a map that considers the ensemble of these units. These maps were drawn in *ArcGIS Map* based on published stratigraphic thicknesses, well data, estimated thickness from geological maps, and seismic profiles. The source of each data point (Injana Formation: 42; Mukdadiya Formation: 33) is available in the supplementary data section (S1). For the Fatha Formation, the construction of

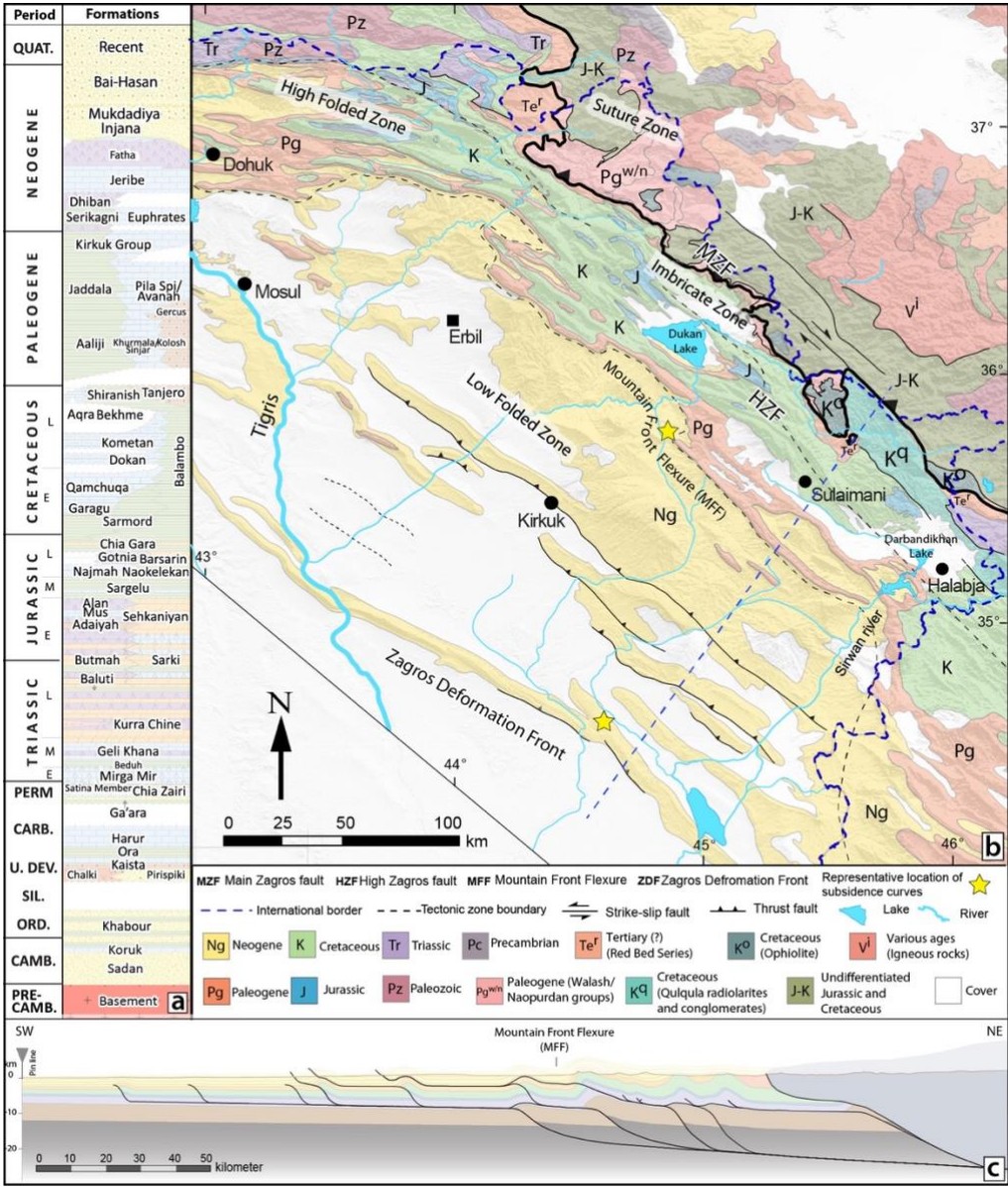

**Figure 2: (a)** Chronostratigraphic column showing the major rock units in the NW Zagros belt, including the Neogene Zagros foreland basin deposits (Fatha, Injana, Mukdadiya, and Bai-Hasan formations). After English et al. (2015). **(b)** Geologic map of the NW Zagros fold-thrust belt and foreland basin exhibiting the spatial distribution of key rock exposures and structural subdivisions (Koshnaw et al., 2020a), **(c)** Balanced cross-section across the NW Zagros fold-thrust belt and foreland basin (see the geologic map for location) highlighting the key subsurface faults and fold-thrust belt geometry (Koshnaw et al., 2020a).

the isopach map was guided by the published contour lines of Dunnington (1958). The other maps were drafted solely relying on the control points.


Subsidence curves derived from backstripping were established for two localities (Fig. 1b): one in the north adjacent to the MFF, near the hinterland, and another one in the south in the vicinity of the depocenter. This was accomplished using the method described in Angevine et al. (1990). Data on thicknesses, lithologies, densities, porosities, compaction coefficients, and water depth were compiled from Al-Naqib (1959), Al-Sheikhly et al. (2015), Sachsenhofer et al. (2015), Koshnaw et al.

(2020b). The data used for creating the subsidence curves are available in the supplementary data section (S2).

The flexural response of the foreland plate to loading was calculated using the concept of a broken beam overlying a fluid substratum using *Flex2D* (v. 5.2: Cardozo, 2021), which employs solutions based on Hetenyi (1946) and Bodine (1981). We used variable elastic thicknesses below the tectonic load (30-50 km from the MZF to the MFF) and a constant elastic thickness

below the sedimentary load (50 km beyond the MFF). The elastic thickness variation was estimated based on Saura et al. (2015) and Pirouz et al. (2017). A balanced cross-section (Koshnaw et al., 2020a) across the NW Zagros fold-thrust belt and foreland basin was used as a base for the modeling. Loads were measured above a horizontal reference line projected mountainward from the bottom of the undeformed Fatha Formation in the foreland (pin line location). For the purpose of flexural modeling, the trace of the balanced cross-section was extended to western Iraq, where the Fatha Formation and older

rock units are exposed. Input data and used parameters can be found in the supplementary data section (S3).

Data for constructing the Bouguer gravity anomaly map was obtained from the Earth Gravitational Model (EGM2008: Pavlis et al., 2008), which has a spatial resolution of 2.5 arc-minute by 2.5 arc-minute (~3.7 km at latitude N36°). The dynamic topography map was digitized from Craig et al. (2011: Fig. 15d), which was calculated from the free-air gravity anomaly.

Crystallization ages of igneous rocks with mantle origin were taken from Ball et al. (2021) to assess magmatic activity. Tomographic maps and profiles are based on the global P-wave velocity anomaly model UU-P07 by Amaru (2007) and Hall, and Spakman (2015) and processed using the web-based tool *SubMachine* (Hosseini et al., 2018). The spatial resolution of the tomographic image varies depending on depth: 50 km from the base of the crust down to 410 km and then 65 km down to 660 km (Hall and Spakman, 2015). P-wave velocity anomalies are indicated relative to the reference model ak135 of Kennett et al.

180  (1995).

## 4 Results

### 4.1 Isopach maps

The isopach maps for the Fatha, Injana, and Mukdadiya formations display variations in the basin architecture and illustrate an overall southeastward increase in stratigraphic thicknesses (Fig. 3). The map for the Fatha Formation is noticeably different

from those of the Injana and Mukdadiya formations in the sense that the isopach pattern suggests a depocenter with c. 600 m thick sediments parallel to the NW-SE trend of the Zagros fold-thrust belt. By contrast, the thickness of the overlying Injana and Mukdadiya formations points to a southeasternward shift in the basin axis, which also rotated to a more N–S orientation. The thickness of the Fatha, Injana, and Mukdadiya formations collectively reaches ~3500 m in the southeastern segment of the study area.


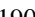

**Figure 3: Isopach maps for the period ~20-5 Ma illustrating the thickness and depozone's distribution for the Fatha (a), Injana (b), Mukdadiya (c) formations, and all together (d). Note the basin geometry change after the deposition of the Fatha Formation (middle Miocene). Black dots represent control points used to construct the contour lines. The Fatha Formation isopach map is based mainly**
**on Dunnington (1958). The black dashed line in d represents the profile location used in flexural modelling (see Fig. 5).**

## 4.2 Subsidence curves and forward modeling of the flexural profile

The subsidence curves obtained from backstripping (see the yellow stars in Figure 2 for location of curves) show an overall convex-up shape, which is mainly characterized by rapid tectonic subsidence after ~20 Ma (Fig. 4), consistent with other examples of foreland basin subsidence curves (e.g., Xie and Heller, 2009). Flexural modeling of the Neogene Zagros tectonic and sedimentary loads (Fig. 5) demonstrates that both loads produce a shallower accommodation than the observed depth of the Neogene basin (base of Fatha Formation).

For the northeastern part of the basin near the MFF (Fig. 2), the subsidence curve exhibits a convex-up shape between ~80-60 Ma and later after ~40 Ma, followed by a ~25 Ma unconformity (Fig. 4: the grey curve labelled as mountain front). The southern subsidence curve (Fig. 4: the green curve labelled as foredeep) involves an unconformity between ~72 and 60 Ma, and shows slow subsidence for most of the Paleogene.

Overall, the tectonic load is responsible for ~1100-1300 m subsidence in the NW Zagros belt (Figure 5), and the tectonic load alone could produce a deflection as deep as ~1387 m near the mountain front flexure (MFF), which is comparable to the tectonic subsidence resulting from backstripping (Fig. 4). Near the MFF, the modelled tectonic and sedimentary loads together leads to ~2500-2000 m of subsidence, yet in the depocenter of the basin, where the observed depth is ~3500-3000 m, the predicted deflection depth due to tectonic and sedimentary loads is only ~2000-1500 m. Accordingly, the modelled basin subsidence from tectonic and sedimentary loads is less than the total accommodation that was generated in the basin by ~800-600 m since the early Miocene when the deposition of the Fatha Formation commenced (Figure 5).

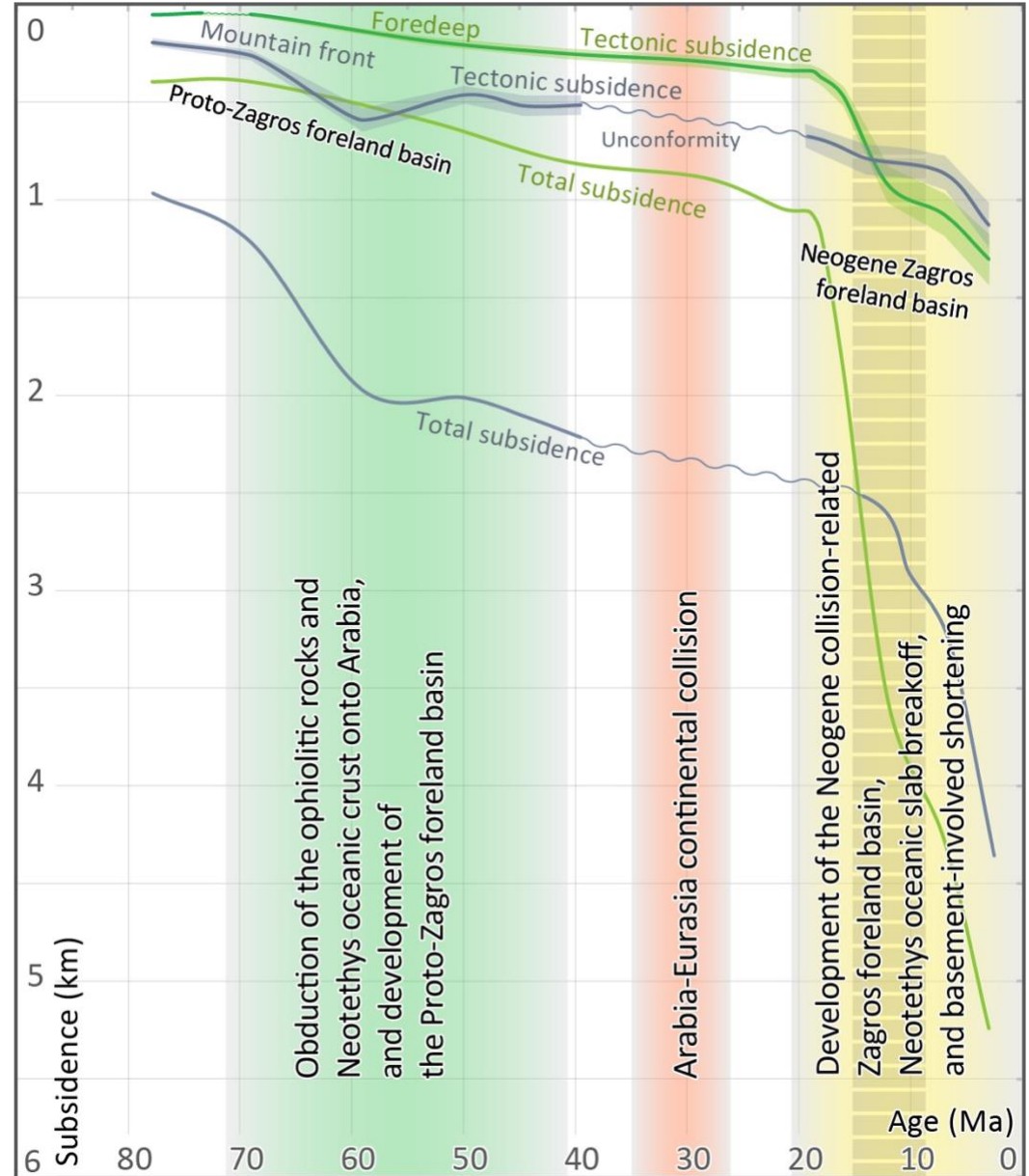

**Figure 4: Subsidence analysis for two locations (mountain front in purple and foredeep in green; see estimated locations in Fig. 3d) showing tectonic and total subsidence. The shaded color of the tectonic subsidence curves depicts a 10% error. Colored and grey dashed bars represent major tectonic events in the geologic record of northeastern Arabia.**

220

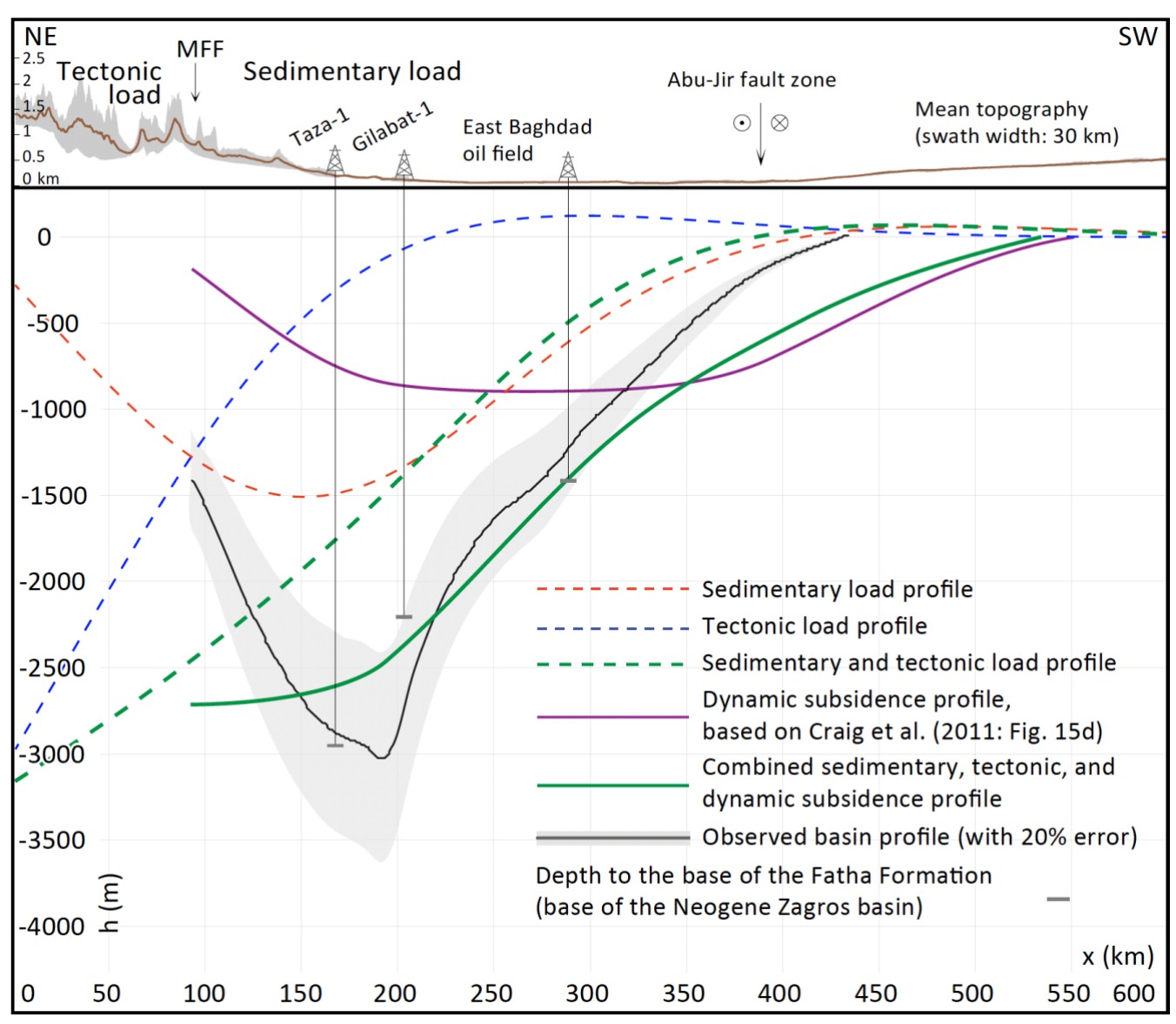

**Figure 5: Modeled flexural profiles due to sedimentary and tectonic loads, and both loads together, as well as their combination with the dynamic subsidence profile in comparison to the observed basin flexure across the southeastern segment of the study area (see the trace line in Fig. 2b). Dynamic subsidence curve is based on the dynamic topography map of Craig et al. (2011: Fig. 15d). The observed basin profile is the arithmetic summation of the constructed isopach maps. MFF–mountain front flexure.**

## 4.3 Bouguer gravity anomaly, dynamic topography, and tomography maps

The observed regional Bouguer gravity anomaly in the NW Zagros fold-thrust belt and foreland basin highlights a notable difference between the NE/E and the SW/W (Fig. 6a). In this sense, the overall negative gravity values in the NE/E may represent a deeper basin and the gravity signal of a crustal root beneath the belt. In the dynamic topography map (Fig. 6b), the positive values point towards the occurrence of uplift along the Arabia–Eurasia suture zone in the NE/E and most of the northeastern fold-thrust belt area. In contrast, the foreland basin part is dominated by subsidence, which reaches ~800 m in the southeast. Moreover, the present-day spatial organization of the Tigris and its tributaries, overall, coincides with the variation in the dynamic topography, but locally, it is more influenced by basement strike-slip faults, where the Bouguer gravity anomaly values show changes.

The tomographic maps (200 km depth slice) of the Middle East (Fig. 7a) and the tomographic profiles across the NW Zagros fold-thrust belt and foreland basin (Fig. 7b) show the occurrence of a N-S-oriented zone of a low seismic velocity (presumably hot material with a lower density) anomaly in western Arabia and eastern Turkey, and distribution of high velocity (high-density cold material) anomalies beneath the NW Zagros belt (Fig. 7a,b). The tomographic profiles I, II, and III (Fig. 7b) indicate that low-density hot material dominates the upper ~400 km of the upper Mantle in northern Arabia and eastern Turkey, whereas farther southward, high-density cold material is prominent throughout the upper mantle. Furthermore, the crystallization age of mantle magmatism since the late Eocene (Ball et al., 2021) plotted on the tomography maps shows a spatial correlation with the distribution of the hot mantle material that is also characterized by a low seismic velocity (Fig. 7c).

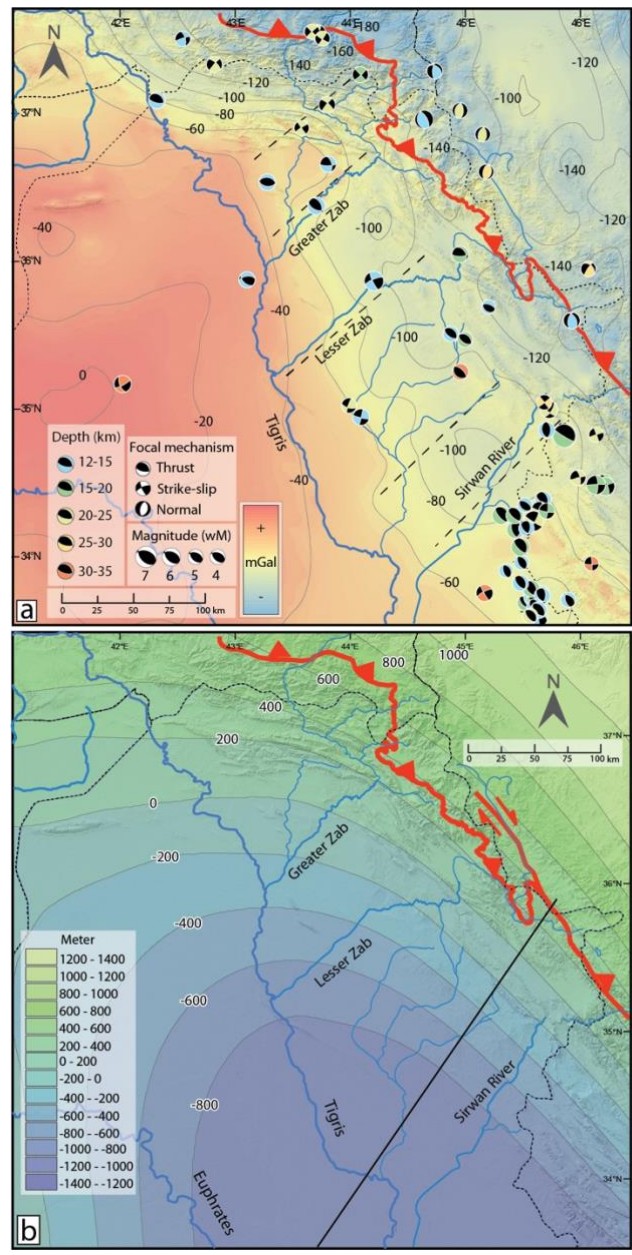

**Figure 6: (a) Regional Bouguer gravity anomaly map with a spatial resolution of 2.5 arc-minute by 2.5 arc-minute (EGM2008: Pavlis et al., 2008) for the NW Zagros fold-thrust belt and foreland basin, showing low values in the northeastern segment of the basin discontinued by strike-slip faults. Earthquake focal mechanisms are from 1976 to 2019, using the Global CMT database (Dziewonski et al., 1981; Ekström et al., 2012). (b) Dynamic topography map of the NW Zagros fold-thrust belt and foreland basin that digitized from the published map of Craig et al. (2011: Fig. 15d). The black line in b represents the profile location used in Fig. 5.**

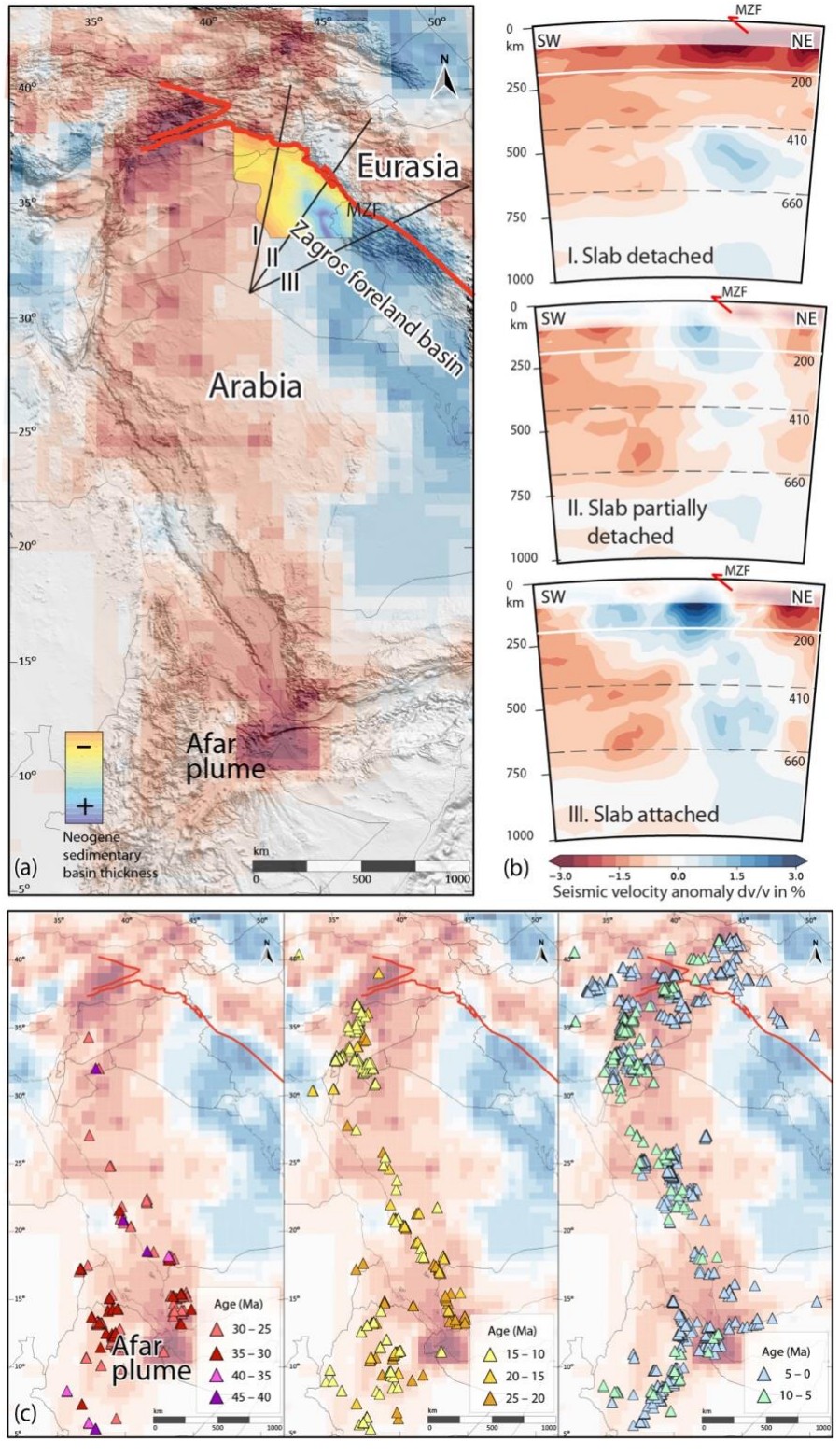

**Figure 7: (a) Neogene deposits' isopach map (~20-5 Ma) of the NW Zagros foreland basin posted on the tomographic and shaded relief maps, illustrating that the basin depocenter is where the high-density cold upper mantle material is and vice-versa. (b) Tomographic cross-section across lines I, II, and III across the NW Zagros fold-thrust belt and foreland basin. From top to bottom (NW to SE), the northeasterly vergence subducted high-velocity slab shows detachment to attachment. MZF–Main Zagros fault. (c) Tomographic map slices at 200 km depth of the Middle East based on the global P-wave velocity anomaly model UU-P07 (Amaru, 2007; Hall and Spakman, 2015) constructed by using SubMachine (Hosseini et al., 2018). From left to right, based on the crystallization age of mantle magmatism (Ball et al., 2021), magmatic rocks are subdivided into 45-25 Ma, 25-10 Ma, and 10-5 Ma and plotted on the tomographic map. The spatial distribution of magmatism through time is in harmony with the upper mantle longitudinal N-S low seismic velocity (low-density hot material), highlighting the influence of the Afar plume generation and northward flow.**

## 5 Discussion

### 5.1 Flexural subsidence in the NW Zagros foreland basin

Lines of evidence from this study reveal the NW Zagros basin genesis as a flexural foreland basin that was later, during the late Miocene, influenced by dynamic topography due to slab tearing in the NW and northward mantle flow. The shift in stratigraphic thicknesses, displayed by the isopach maps since the deposition of the Fatha Formation, indicates that the basin continuously subsided in the southeastern part of the study area. In contrast, the northern and northwestern parts experienced a reduction in the generation of accommodation during the deposition of the Injana, Mukdadiya, and Bai-Hasan formations. During the early Miocene, when the Fatha Formation was deposited, flexural bending due to the combined effect of topographic and slab loads was the primary mechanism for driving the subsidence of the Arabian foreland plate. We infer that this mechanism was at work because of the NW–SE trending paleotopography of the proto-Zagros fold-thrust belt, forming a topographic load, and the subduction of the Arabian plate, resulting in a bending component due to slab load. Afterward, the NW Zagros foreland basin differentially subsided, resulting in a deep basin in the southeastern part. As presented in the section on the results of the flexural modeling, the present-day topography cannot produce a basin flexure as deep as the observed depth. The incompatibility between the observed and predicted flexural profiles was also proposed by Saura et al. (2015). However, another flexural profile model (Pirouz et al., 2017) predicts that the observed basin profile is shallower in the NW Zagros belt, contrary to the presented model in this study. Such inconsistency could be related to the fact that the latter model (1) considered the top of the Asmari Formation as the base of the foreland basin, and (2) the limit of the topographic load extended beyond the Arabia-Eurasia suture. In this study, the base of the Fatha Formation was considered as the base of the foreland basin based on the results of the subsidence curve analysis (Fig. 4). Additionally, the drainage divide, which spatially coincides with the MZF, was considered as the limit of the topographic load (Sinclair and Naylor, 2012).

The isopach map configuration of the Injana Formation hints at an orogen-parallel river system (Fig. 3b), whereas the Mukdadiya Formation map points to sediment input from an orogen-perpendicular river into the existing orogen-parallel system (Fig. 3c), in line with results from provenance data. The synthesis of provenance data, accomplished on the basis of

detrital zircon U-Pb ages from the Injana to the Mukdadiya formations (Koshnaw et al., 2020b: Fig. 10), shows a noticeable increase in the Paleogene age component (from ~20% to ~50%), pointing to a sediment source situated in the eastern terranes. Conversely, the Paleozoic age components, which point to material sources situated mainly in the north and northwestern terranes, decreased during the same time (from ~50% to ~30%). Such provenance change suggests an earlier uplift in the north and northwestern terranes during the deposition of the Injana Formation compared to the eastern terranes that were uplifted mostly during the deposition of the Mukdadiya Formation. This inference is also consistent with the synthesis of the drainage network evolution in the Euphrates and Tigris river basin (Wilson et al., 2014; McNab et al., 2017), which points to the occurrence of an earlier topographic growth in the northern and northwestern terranes rather than in the southern or southeastern segments of the Zagros orogen. Accordingly, provenance data in combination with information on the growth history of the topography in the source area suggest the occurrence of terrane uplift in the northwest at the early stage of the foreland basin paired with a rapidly subsided basin in the southeast, even though the southeastern terranes were not uplifted significantly. Therefore, flexural subsidence alone is unlikely to explain the deep basin in the southeastern part of the study area, and hence, an additional driving force is necessary.

**5.2. Northward flow of the Afar asthenospheric mantle plume, lithospheric slab tearing, and dynamic topography**

Dynamic subsidence in combination with flexure subsidence could be invoked to conjointly account for the observed ~3-4 km thick accommodation in the southeastern segment of the basin (Fig. 5). Deep earth processes such as horizontal slab tearing beneath the northwestern segment of the Arabia-Eurasia collision zone and the Afar mantle plume flow above the detached segment could be linked to the localization of the basin subsidence in northeast Arabia and uplift in the northwestern segment. Slab-pull force is a crucial force driving the motion of a plate, and this mechanism can be altered by slab tearing and detachment (Wortel and Spakman, 2000). In regions where the slab is detached, the slab-pull force no longer impacts the surface above the detachment, but instead, the force intensifies on the still attached part of the slab. This change in the force balance results in an enhanced downward tension on the tearing tip and in a lateral propagation of the detachment. The surface region above the detached segment of the slab undergoes uplift, whereas the area above the still attached slab experiences subsidence (van der Meulen et al., 1998; Wortel and Spakman, 2000). These predictions are observed in the evolution of the NW Zagros basin.

Tectonic subsidence curves constrain the onset of foreland basin subsidence to the timing of the Fatha Formation deposition. The reported depositional time of the Fatha Formation spans from the early to middle Miocene as constrained by the $^{87}Sr/^{86}Sr$ isotope chronostratigraphy and the fossils content (Al-Naqib, 1959; Mahdi, 2007; Grabowski and Liu, 2010; Hawramy and Khalaf, 2013). In the vicinity of the southeastern segment of the study area, the Fatha Formation samples that were analyzed for the $^{87}Sr/^{86}Sr$ isotope indicate an age of ~15.5 Ma near the contact with the Injana Formation (Grabowski and Liu, 2010). Later, in the middle-late Miocene, during the deposition of the Injana Formation, the basin underwent rapid subsidence in the SE, but the NW segment experienced the genesis of limited accommodation (Figs. 3). Furthermore, the Bouguer gravity anomaly map suggests a remarkable difference in basement depth between the northeast, and the north and southwest of the

NW Zagros belt (Fig. 6a). This variation of depth was interpreted to be as shallow as ~5-6 km in the west and as deep as ~12

330 km in the east (Konert et al., 2001). The shallower depth-to-basement in the western part of the NW Zagros belt may also be related to the variation in the paleotopography of the basement. In the NW and W Iraq, paleohighs with reduced sedimentation have been suggested for periods even before the Miocene (Jassim and Buday, 2006). However, considering how consistent the isopach map of the Fatha Formation throughout the basin is, potential paleohighs seem to have had a limited or no influence on the low accommodation during the deposition of the post-Fatha formations. We advocate that this variation of the basin

subsidence in the east versus west after the early Miocene is linked to the slab tearing and dynamic subsidence. As estimated from free-air gravity anomaly (Craig et al., 2011), dynamic subsidence dominates the area in the NW Zagros belt (Fig. 6b). It results in the deflection of the southeastern segment of the basin up to ~800 m. Combining the amount related to the dynamic subsidence with that of the flexural subsidence (due to surface load) successfully reproduces the observed ~3-4 km depth of the southeastern segment of the NW Zagros basin (Figs. 5), arguing for mantle influence (Fig. 7).

The lithospheric process of horizontal slab tear actively induces and reorders the mantle flow, leading to a long wavelength variation in the surface topography (Dávila and Lithgow-Bertelloni, 2013; Faccenna et al., 2013). In the northern Middle East, Neotethys oceanic slab detachment and hot mantle material emplacement at shallow depth have been suggested between the Arabian and Eurasian plates and beneath the Turkish–Iranian plateau (TIP) on the basis of geochemical data and seismic

tomography (Keskin, 2003; Hafkenscheid et al., 2006; Omrani et al., 2008; Koulakov et al., 2012; Kaviani et al., 2018). On the surface (Figs. 7a,b), lateral migration of detachment and the associated shift in the mantle flow are manifested by uplift where the slab is detached due to rebounding and the replacement by hot mantle material, and further subsidence where it is still attached due to slab weight and the downward flow of the mantle material. The northward spatial distribution of the Afar plume mantle magmatism since the middle Eocene (Camp and Roobol, 1992; Ershov and Nikishin, 2004; Faccenna et al.,

2013) is suggestive of a ~15-10 Ma slab detachment. This is inferred from the occurrence of mantle magmatism beyond the Arabia-Eurasia suture zone only after ~10 Ma (Figs. 7c, 8). The occurrence of a northward flow of the Afar mantle material is further supported by thickness anomaly in the mantle transition zone (Kaviani et al., 2018) and data on the orientation of the fast velocity polarization inferred from SKS splitting (Faccenna et al., 2013). This northward flow of the Afar plume is thought to have influenced the Mediterranean tectonic history and contributed to the Arabia-Eurasia convergence (~3-2 cm/yr), even

after the Oligocene continental collision (Jolivet and Faccenna, 2000; Alvarez et al., 2010; Boutoux et al., 2021) (Fig. 9). Additionally, it is also manifested by the inversion of the right-lateral Abu-Jir fault zone during the early to middle Miocene in western Iraq, concurrent with the Afar plume mantle material arrival (Figs. 1, 9b-e), away from the area in the east where the Zagros-related shortening was ongoing (Alhadithi et al., 2023).

Taking into account that the collision between northern Arabia and Eurasia occurred during the early Oligocene, slab mechanical weakening and necking were already underway before the arrival of the Afar asthenospheric mantle material during the middle Miocene. Three-dimensional analogue models simulating the Arabia-Eurasia collision suggest that slab tearing

began in northwestern Arabia due to the lateral transition of the Arabian continental crust into the oceanic crust toward the Mediterranean region. This transition led to the subduction of the oceanic plate at the Eurasian margin while the continental plate collided, due to buoyancy differences (Faccenna et al., 2006). The oblique convergence between Arabia and Eurasia (McQuarrie et al., 2003; Navabpour et al., 2008) may have further promoted slab tearing, as this obliquity caused an earlier continent-continent collision compared to adjacent regions where oceanic subduction continued, as demonstrated by numerical models (Boonma et al., 2023). During the middle Miocene, the arrival of the hot asthenospheric mantle likely accelerated slab tearing by thermally weakening the slab necking zone, reducing its viscosity and strength (Keskin, 2007; Menant et al., 2016; Boutelier and Cruden, 2017). By the late Miocene, as the southeastward horizontal tearing of the slab progressed, the northward flow of the Afar plume asthenospheric mantle material advanced into eastern Anatolia. This process triggered a dynamic uplift along western Arabia and the TIP, accompanied by dynamic subsidence on the eastern side (e.g., Daradich et al., 2003; Craig et al., 2011). Additionally, as the Neotethys oceanic plate remained attached to the Arabian continental plate on its eastern side, several numerical models indicate that mantle downward flow along the subducting slab also contributes to the dynamic subsidence (e.g., Bottrill et al., 2012; Duretz and Gerya, 2013; Balázs et al., 2022).

Furthermore, when slab breakoff occurs, the geochemical properties of igneous rocks are anticipated to transition from primarily calc-alkaline (Ca-rich) to more alkaline (Na- and K-rich), accompanied by a shift in εNd values from negative to positive. This geochemical shift is attributed to the cessation of slab-derived fluids following slab breakoff. Similar patterns have been observed in eastern Turkey, where alkalinity has increased since the middle Miocene from north to south (Keskin, 2003; Şengör et al., 2008). In northwestern Iran, the geochemical compositions of the late Miocene-Quaternary Sablan, Sahand, and Saray volcanoes, which are located across the Arabia-Eurasia suture zone from northeast to southwest, respectively, in addition to slab subduction signatures, show medium to very high K content (ultrapotassic), positive εNd values (Sablan), and possibility of hot asthenospheric inflow (Saray) driving partial melting of an already metasomized mantle wedge (Moghadam et al., 2014; Ghalamghash et al., 2019; Chaharlang et al., 2023). Additionally, Miocene magmatism from northwest to southeast, from SE Turkey to NW Iran, shows εNd values shifting from positive to negative (Grosjean et al., 2022), suggesting that the magmas in SE Turkey originated from a primitive mantle melt (slab detached), whereas those in NW Iran resulted from a crustal contaminated melt (slab partially detached), aligning with the hypothesized southeastward slab tearing proposed in this study (Figs. 7a,b, 10).

Along the Tethyan realm, the effects of the lithosphere dynamics on the surface geology have been documented in the (i) Alpine Molasse basin, which led to a change in basin stratigraphy from the Flysch stage to the Molasse stage (Sinclair, 1997; Schlunegger and Kissling, 2022), (ii) the Apenninic basin that controlled foredeep migration (van der Meulen et al., 1998), and in the (iii) Mediterranean region, where the eastern part is influenced by slab detachment, which in turn resulted in the switching of volcanism's geochemical character from calk-alkaline to alkaline (Wortel and Spakman, 1992). To the southeastern frontier of the Zagros orogenic belt, north of the Makran accretionary complex, the Karvandar basin adjacent to

the South Sistan Suture Zone (SE Iran) contains 3.5 km shallow-marine to nonmarine rocks. The Karvandar basin is interpreted to represent a peripheral foreland that underwent a renewed phase of subsidence ~10–15 Myr after the Sistan Suture Zone development, possibly due to slab rollback of the downgoing plate and lithospheric mantle delamination of the overriding plate

(Mohammadi et al., 2016; Ruh et al., 2023).

In short, the lines of evidence from the Zagros belt argue that the NW Zagros foreland basin transitioned from a flexural foreland basin to a mantle–influenced basin due to slab tearing (Fig. 10), which enhanced and localized the occurrence of subsidence in the southeastern part of the basin. Such a pattern in the basin's evolution is not unique to the Zagros belt, and

405 therefore, the provided synthesis of the geologic datasets in this study could be helpful to further constrain subsidence mechanisms in other basins in the world.

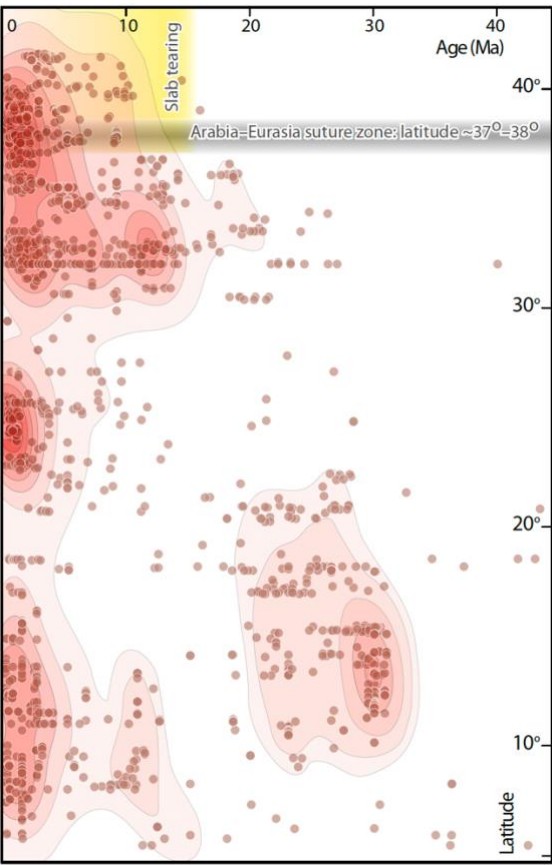

**Figure 8: Mantle-related magmatic rock age versus latitude plotted as bivariate kernel density estimation showing intense**
**magmatism beyond the Arabia-Eurasia suture zone in the Turkish-Iranian plateau after ~10 Ma. Magmatic age data are from Ball et al. (2021).**

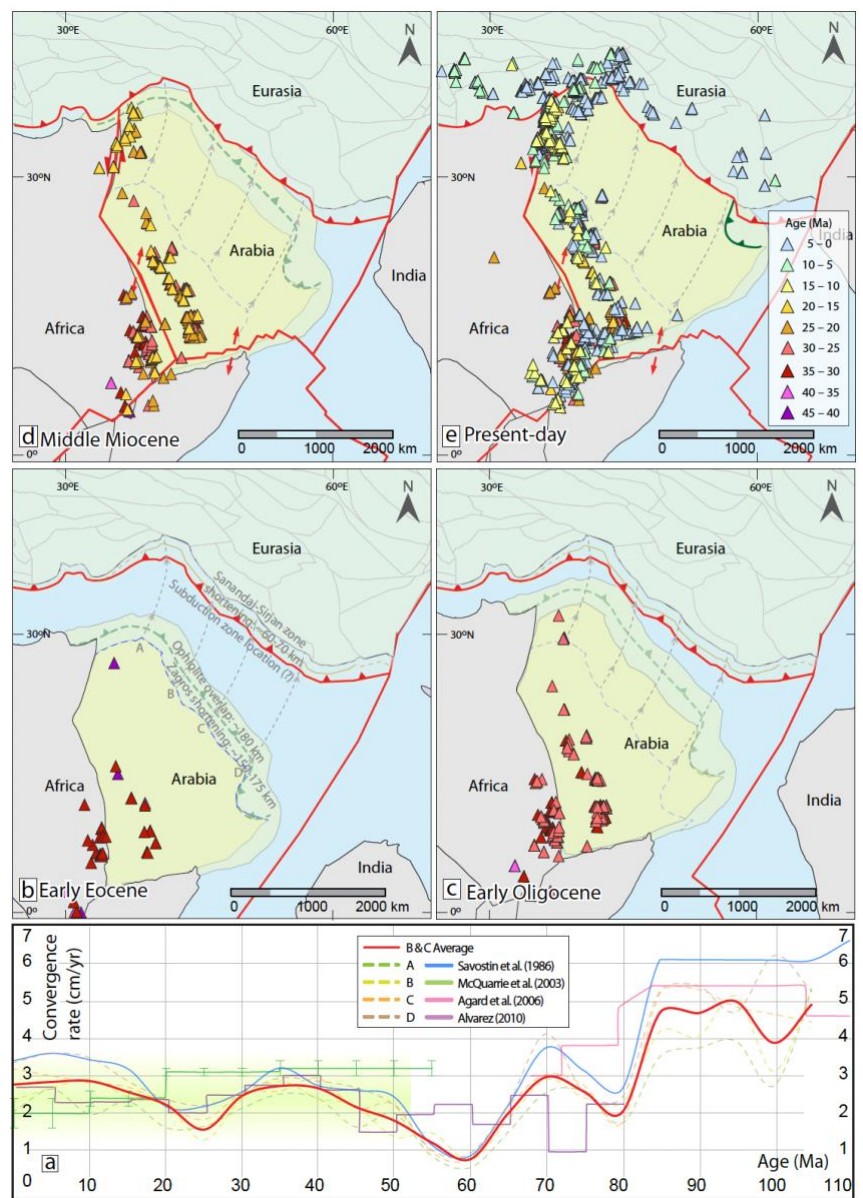

Figure 9: (a) Convergence rate between moving Arabia against fixed Eurasia calculated from reference points (A, B, C, and D) from the Arabian plate based on plate kinematic reconstruction of Müller et al. (2018; GPlates 2.2.0) and compared with other published convergence rate (Savostin et al., 1986; McQuarrie et al., 2003; Agard et al., 2006; Alvarez et al., 2010). (b-e) The Arabian plate kinematic reconstruction with respect to the fixed Eurasia since the early Eocene (~55 Ma) is based on Müller et al. (2018). The width of the Eurasian outer margin during Paleogene is estimated from the present-day width of the Makran accretionary wedge (~500-300 km; e.g., Farhoudi and Karig, 1977).

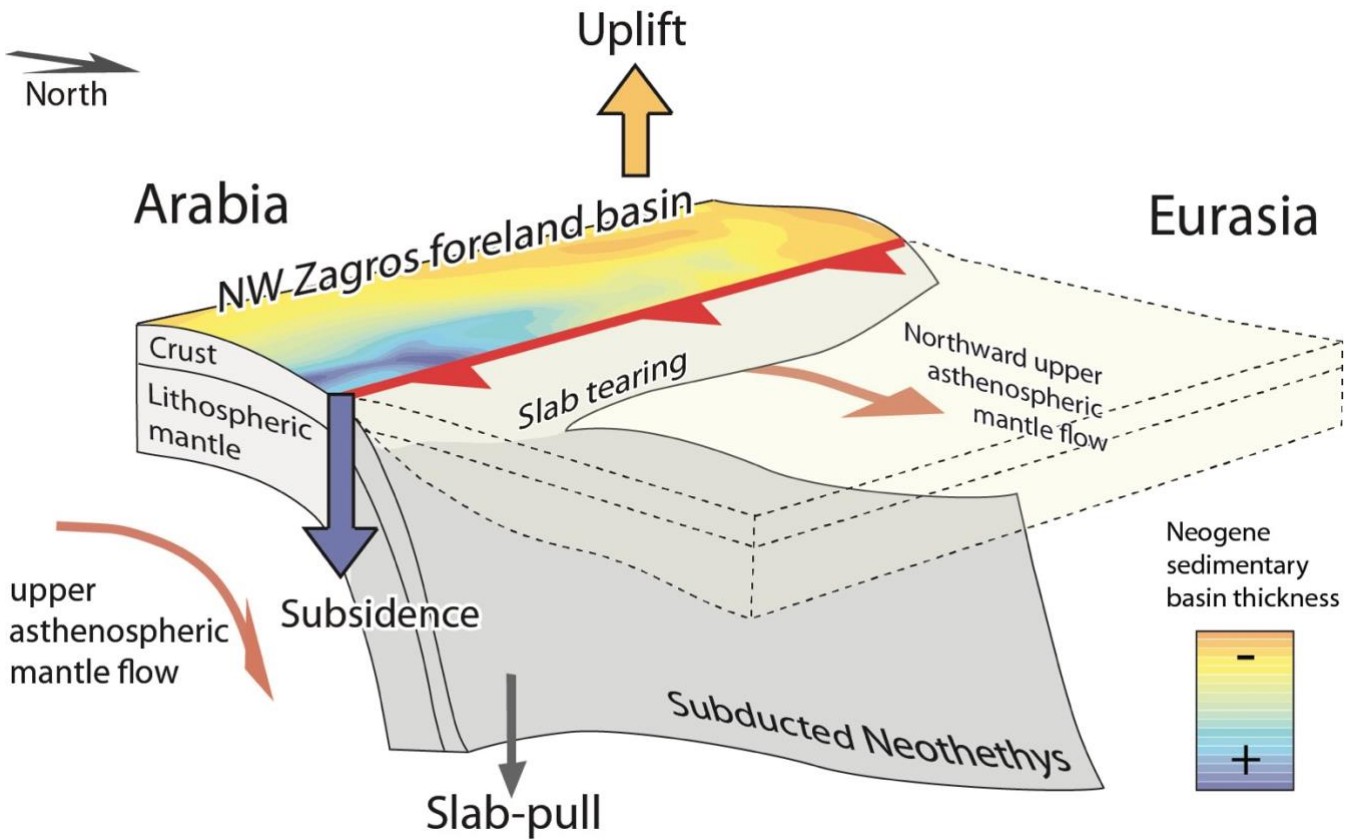

**Figure 10: Conceptual illustration portraying the NW Zagros foreland basin evolution during Neogene in response to the downgoing Neotethys oceanic slab tearing, detachment, and breakoff along the Arabia-Eurasia suture zone from the NW to the SE, and upper mantle material northward flow in western Arabia.**

## 6 Conclusions

In this study, we invoke the combined effects of flexural subsidence and dynamic subsidence to explain the abnormally high accommodation space in the southeastern segment of the Neogene Zagros foreland basin in the Kurdistan region of Iraq. In its southeastern segment, the Neogene Zagros foreland basin thickens to ~3-4 km, incompatible with the adjacent mountain topography (~1.5-2.5 km). Isopach maps for the Miocene formations in the NW Zagros foreland basin reveal the occurrence of continuous subsidence in the southeastern segment, while the northwestern part experienced the generation of limited accommodation space. Flexural subsidence modeling and tectonic subsidence curve analysis, supported by dynamic topography, Bouguer gravity, and seismic tomography maps, show that the present-day Neogene NW Zagros basin depth and geometry can be explained by adding the effect related to dynamic subsidence.

Lines of evidence from this study suggest that the NW Zagros foreland basin underwent a two-stage basin evolution: (1) an early stage of basin flexure due to surface and slab loads during the early Miocene, and (2) a later stage when the basin was modified by dynamic topography due to the propagation of the Neotethys horizontal slab-tear from the northwest to the southeast along the Arabia-Eurasia suture zone. As the Neotethys oceanic crust slab tear propagated SE-ward, the Afar mantle plume flowed beyond the Arabia and Eurasia suture zone in the NW, and the flow distributed beneath the TIP. Based on the timing of mantle magmatism and their spatial distribution beyond the suture zone in the NW, in concordance with the dissemination of the hot mantle, as well as the timing of the shift in basin architecture, the slab tear is expected to have taken place during the middle Miocene. Accordingly, this study demonstrates that the NW Zagros basin's development is a product of both surface and deep earth processes. The interplay between flexural subsidence and dynamic subsidence, driven by horizontal slab tearing and mantle flow, provides a mechanism for understanding the NW Zagros basin's stratigraphic and tectonic evolution after the Arabia-Eurasia collision.

*Data availability.* The data used in this article will be available at Mendeley Data repository and are already available to the reviewers.

*Supplement.*

*Author contributions.* RK: leadership responsibility, funding acquisition, conceptualization, data curation, formal analyses, investigation, visualization, and writing (original draft preparation). Jonas Kley: resources, conceptualization, and writing (review and editing). FS: conceptualization and writing (review and editing).

*Competing interests.* The authors declare that they have no known competing financial interests or personal relationships that could have appeared to influence the work reported in this paper.

*Acknowledgments.* The first author, R. I. Koshnaw, benefited from discussions with M. Zebari, M. Tamar-Agha, M. Kloecking, and M. Correa. This work was fully supported by the Alexander von Humboldt research fellowship awarded to R. I. Koshnaw.

*Financial support.* This work was fully supported by the Alexander von Humboldt research fellowship awarded to R. I. Koshnaw.

*Review statement.*

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
