# Peer review of "The Miocene subsidence pattern of the NW Zagros foreland basin reflects SE-ward propagating tear of the Neotethys slab"

_EGUsphere, 2023_

## Referee Comment (RC1)

The manuscript "Miocene evolution of the NW Zagros foreland basin reflects SE-ward propagating tear of the Neotethys slab" by Koshnaw, Kley, and Schlunegger provides a set of new and important data about the Miocene foreland basin of the Iraqi part of Zagros Mountain range. The manuscript is well written and the multi-proxy data including isopach maps, subsidence curves, flexural profile modeling along with Bouguer gravity anomaly, tomography maps, and dynamic topography data, support the author's findings and their conclusions.

Adding a new chapter "Regional Tectonic Implications" and comparing and discussing some of the other foreland basins in the Tethyan belt makes it interesting for a wider community working on the collisional system of the Tethyan orogenic belt. For example, the Geological evolution of the South Sistan Basin (Sistan Suture Zone) and its Miocene foreland Basin (Karvandar Basin) in SW Iran are very similar to the author's studies foreland basin. The width of the foreland basin and sediment thickness increased toward the south and similar asthenosphere flow after the final collision occurred in the South Sistan Basin. Unlike Zagros, there are limited studies in the Sistan Basin. However, I think the brief discussion of similarities between Zagros and the Sistan Foreland Basin (Karvandar Basin) would further support the author's findings. These two papers would be helpful in this case.

Mohammadi, A., Burg, J.P., Bouilhol, P. and Ruh, J., 2016. U–Pb geochronology and geochemistry of Zahedan and Shah Kuh plutons, southeast Iran: Implication for closure of the South Sistan suture zone. Lithos, 248, pp.293-308.

Ruh, Jonas Bruno, Luis Valero, Mohammad Najafi, Najmeh Etemad-Saeed, J. Vouga, Ali Mohammadi, Fabio Landtwing, Marcel Guillong, Miriam Cobianchi, and Nicoletta Mancin. "Tectono-Sedimentary Evolution of Shale-Related Minibasins in the Karvandar Basin (South Sistan, SE Iran): Insights From Magnetostratigraphy, Isotopic Dating, and Sandstone Petrology." Tectonics 42, no. 11 (2023): e2023TC007971.

In addition, please instead of review papers, cite the original papers. For example, in the caption of Fig. 9 **"present-day width of the Makran accretionary wedge (~500-300 km; e.g., Burg et al. 2018)"** Please cite:

McCall, G.J.H., 1997. The geotectonic history of the Makran and adjacent areas of southern Iran. Journal of Asian Earth Sciences, 15(6), pp.517-531.

**Or** Farhoudi, G. and Karig, D.E., 1977. Makran of Iran and Pakistan as an active arc system. Geology, 5(11), pp.664-668.

Some recent papers about the timing of the collision support the Late Oligocene Arabia-Eurasia continental collision. It's worth citing them too. For example:

Cai, F., Ding, L., Wang, H., Laskowski, A.K., Zhang, L., Zhang, B., Mohammadi, A., Li, J., Song, P., Li, Z. and Zhang, Q., 2021. Configuration and timing of collision between Arabia and Eurasia in the Zagros collision zone, Fars, southern Iran. Tectonics, 40(8), p.e2021TC006762.

---

## Author Comment (AC1)

RC1: 'Comment on egusphere-2023-3123', Ali Mohammadi, 11 Feb 2024

The manuscript "Miocene evolution of the NW Zagros foreland basin reflects SE-ward propagating tear of the Neotethys slab" by Koshnaw, Kley, and Schlunegger provides a set of new and important data about the Miocene foreland basin of the Iraqi part of Zagros Mountain range. The manuscript is well written and the multi-proxy data including isopach maps, subsidence curves, flexural profile modeling along with Bouguer gravity anomaly, tomography maps, and dynamic topography data, support the author's findings and their conclusions.

Adding a new chapter "Regional Tectonic Implications" and comparing and discussing some of the other foreland basins in the Tethyan belt makes it interesting for a wider community working on the collisional system of the Tethyan orogenic belt. For example, the Geological evolution of the South Sistan Basin (Sistan Suture Zone) and its Miocene foreland Basin (Karvandar Basin) in SW Iran are very similar to the author's studies foreland basin. The width of the foreland basin and sediment thickness increased toward the south and similar asthenosphere flow after the final collision occurred in the South Sistan Basin. Unlike Zagros, there are limited studies in the Sistan Basin. However, I think the brief discussion of similarities between Zagros and the Sistan Foreland Basin (Karvandar Basin) would further support the author's findings. These two papers would be helpful in this case.

Mohammadi, A., Burg, J.P., Bouilhol, P. and Ruh, J., 2016. U–Pb geochronology and geochemistry of Zahedan and Shah Kuh plutons, southeast Iran: Implication for closure of the South Sistan suture zone. Lithos, 248, pp.293-308.

Ruh, Jonas Bruno, Luis Valero, Mohammad Najafi, Najmeh Etemad-Saeed, J. Vouga, Ali Mohammadi, Fabio Landtwing, Marcel Guillong, Miriam Cobianchi, and Nicoletta Mancin. "Tectono-Sedimentary Evolution of Shale-Related Minibasins in the Karvandar Basin (South Sistan, SE Iran): Insights From Magnetostratigraphy, Isotopic Dating, and Sandstone Petrology." Tectonics 42, no. 11 (2023): e2023TC007971.

Dear R1 (Ali Mohammadi),

Thank you for your helpful comments. We agree that providing relevant foreland bain examples will enhance the manuscript. In accordance with your suggestion, we added the following text to the discussion section 5.2 (line 620-630):

Along the Tethyan realm, the effects of the lithosphere dynamics on the surface geology have been documented in the Alpine Molasse basin, which led to a change in basin stratigraphy from the Flysch stage to the Molasse stage (Sinclair, 1997; Schlunegger and Kissling, 2022), Apenninic basin that controlled foredeep migration (Meulen et al., 1998), and in the Mediterranean region, where the eastern part is influenced by slab detachment and resulted in the switching of volcanism's geochemical character from calk-alkaline to alkaline (Wortel and Spakman, 1992). To the southeastern frontier of the Zagros orogenic belt, north of the Makran accretionary complex, the Karvandar basin adjacent to the South Sistan Suture Zone (SE Iran) contains 3.5 km shallow-marine to nonmarine rocks. The Karvandar basin is interpreted to represent a peripheral foreland that underwent a renewed phase of subsidence ~10–15 Myr after the Sistan Suture Zone development, possibly due to slab rollback of the downgoing plate and lithospheric mantle delamination of the overriding plate (Mohammadi et al., 2016; Ruh et al., 2023).

In addition, please instead of review papers, cite the original papers. For example, in the caption of Fig. 9

"present-day width of the Makran accretionary wedge (~500-300 km; e.g., Burg et al. 2018)" Please cite:

McCall, G.J.H., 1997. The geotectonic history of the Makran and adjacent areas of southern Iran. Journal of Asian Earth Sciences, 15(6), pp.517-531.

Or  Farhoudi, G. and Karig, D.E., 1977. Makran of Iran and Pakistan as an active arc system. Geology, 5(11), pp.664-668.

 Now "Farhoudi and Karig, 1977" is cited in the caption of Fig. 9

Some recent papers about the timing of the collision support the Late Oligocene Arabia-Eurasia continental collision. It's worth citing them too.  For example:

Cai, F., Ding, L., Wang, H., Laskowski, A.K., Zhang, L., Zhang, B., Mohammadi, A., Li, J., Song, P., Li, Z. and Zhang, Q., 2021. Configuration and timing of collision between Arabia and Eurasia in the Zagros collision zone, Fars, southern Iran. Tectonics, 40(8), p.e2021TC006762.

In the Tectonostratigraphic context section, the new text below is introduced to provide that rationale for the preferred age of collision in the NW Zagros, and the relevant papers are cited.

Various ages for the Arabia-Eurasia collision have been suggested, spanning from the late Cretaceous to the Pliocene (Dewey et al., 1973, 1986; Berberian and King, 1981; Stoneley, 1981; Dercourt et al., 1986; Hempton 1987; Alavi 1994; Agard et al. 2005, 2011; Fakhari et al. 2008; Ballato et al. 2011; Khadivi et al. 2012; McQuarrie and van Hinsbergen 2013; Saura et al. 2015; Gholami Zadeh et al., 2017; Pirouz et al., 2017; Darin and Umhoefer, 2022; Sun et al., 2023). However, as continental collision is a prolonged process, the onset and culmination of the process need to be distinguished. Additionally, the collisional age must agree with well-constrained global paleotectonic models and geologic records. Continental collision is defined by the total subduction of oceanic crust between two continental crusts (Dewey and Horsfield, 1970). In the NW Zagros belt, considering the Arabia-Eurasia collisional age older than Oligocene is less likely due to (1) the necessity for unrealistically long post-collisional subduction of the Arabian continental crust beneath Eurasia (e.g., McQuarrie and Hinsbergen, 2013) and (2) the pervasive occurrence of oceanic crust subduction-related magmatism during Paleocene and Eocene (e.g., Chiu et al., 2013). Considering the Arabia-Eurasia collisional age younger than Oligocene, does not account for lines of evidence from provenance, geochronology, and thermochronology studies, and timing crustal deformation as well as regional tectonostratigraphic observations (Allen and Armstrong, 2008; Koshnaw et al., 2019, 2021; Cai et al., 2021; Song et al., 2023).

---

## Author Comment (AC2)

Based on the isopach maps, subsidence curves, and reconstructions of flexural profiles, supported by Bouguer anomaly data and maps of dynamic topography and seismic tomography, the authors discussed the intricate interplay between deep earth dynamics and surface processes, and then proposed a two-stage evolution of the NW Zagros foreland basin from flexural subsidence to dynamic subsidence. Moreover, the authors suggested that the subsidence of the Zagros foreland basin was triggered by surface loading and slab subduction during the early Miocene, and the tear propagation of the NeoTethys horizontal slab during the late Miocene.

Overall, this manuscript exhibits a high level of organization and clarity in its presentation. I would suggest a middle revision before final acceptance.

General comments:

1. The authors proposed the notion of northward flow of the Afar plume, drawing from the spatial distribution analysis of the crystallization age of mantle magmatism (Ball et al., 2021; Fig. 7). However, it's important to note that the spatial distribution analysis provided may not be comprehensive. There are many Late Eocene mantle magmatic outcrops in Iran (e.g. Deevsalar et al., 2018, JGS), which are missing in Fig. 7c. Therefore the evidence supporting this northward flow remains uncertain.

If the authors maintain their stance on the northward flow of the Afar plume, it is essential that they provide additional evidence to support their claim. This could include a comparative analysis highlighting the similarities and differences between plume-derived magmatism and other types of mantle sources. For instance, it would be valuable to discuss how mantle-derived magmatism resulting from processes like slab break-off, slab tearing, and sub-parallel subduction of the NeoTethys ridge can be distinguished from plume-related magmatism.

Dear R2 (Anonymous),

Thank you for your supportive and constructive comments:

We proposed a northward flow of the Afar plume based on the distribution of mantle magmatism, upper mantle structure as imaged by teleseismic, and fast velocity polarization orientation analysis (SKS splitting) as documented in previous publications (Camp and Roobol, 1992; Ershov and Nikishin, 2004; Faccenna et al., 2013; Kaviani et al., 2018). We agree that further geochemical synthesis will enhance the manuscript, but it may deviate from the focus of our manuscript, which basin analysis.

Indeed the plotting of the 20 (?) samples from Deevsalar et al. (2018) is not included, but the plotted complication from Ball et al., 2021 represents 4443 samples of alkaline and sub-alkaline mafic and ultramafic rocks. Furthermore, the samples from Deevsalar et al. are located in the central part of the Zagros, away from the northwestern segment, which is the focus of this study.

2. The timing of continental collisions is a topic fraught with controversy. While the authors have chosen to adopt the Oligocene time frame in the text and Fig. 4, it would enhance the logical flow of the discussion to provide a comprehensive summary of the various proposed collision times. Furthermore, it would be beneficial for the authors to elucidate the criteria used to select the Oligocene time frame as the preferred collision time, thereby providing readers with insight into the rationale behind this decision.

In the Tectonostratigraphic context section, the new text below is introduced to provide that rationale for the preferred age of collision in the NW Zagros, and the relevant papers are cited.

Various ages for the Arabia-Eurasia collision have been suggested, spanning from the late Cretaceous to the Pliocene (Dewey et al., 1973, 1986; Berberian and King, 1981; Stoneley, 1981; Dercourt et al., 1986; Hempton 1987; Alavi 1994; Agard et al. 2005, 2011; Fakhari et al. 2008; Ballato et al. 2011; Khadivi et al. 2012; McQuarrie and van Hinsbergen 2013; Saura et al. 2015; Gholami Zadeh et al., 2017; Pirouz et al., 2017; Darin and Umhoefer, 2022; Sun et al., 2023). However, as continental collision is a prolonged process, the onset and culmination of the process need to be distinguished. Additionally, the collisional age must agree with well-constrained global paleotectonic models and geologic records. Continental collision is defined by the total subduction of oceanic crust between two continental crusts (Dewey and Horsfield, 1970). In the NW Zagros belt, considering the Arabia-Eurasia collisional age older than Oligocene is less likely due to (1) the necessity for unrealistically long post-collisional subduction of the Arabian continental crust beneath Eurasia (e.g., McQuarrie and Hinsbergen, 2013) and (2) the pervasive occurrence of oceanic crust subduction-related magmatism during Paleocene and Eocene (e.g., Chiu et al., 2013). Considering the Arabia-Eurasia collisional age younger than Oligocene, does not account for lines of evidence from provenance, geochronology, and thermochronology studies, and timing crustal deformation as well as regional tectonostratigraphic observations (Allen and Armstrong, 2008; Koshnaw et al., 2019, 2021; Cai et al., 2021; Song et al., 2023).

Detailed comments:

1. The conclusion section could benefit from some refinement. The authors may consider restructuring it with a series of concise paragraphs following a brief summary to enhance clarity and reader comprehension.

The conclusion section is extensively edited and organized into two paragraphs.

2. The font size in some figures appears to be too small, such as the well names in Fig. 5 and the age legend in Fig. 7c.

The font size in both figures has been updated.

3. The reference list requires updating.

The reference list has been updated.

---

## Referee Report (RR1)

After reading the revised version of the manuscript titled: "The Miocene subsidence pattern of the NW Zagros foreland basin reflects SE-ward propagating tear of the Neotethys slab" by Renas Koshnaw, Jonas Kley, and Fritz Schlunegger, I am happy to let you know the authors responded to my comments, clarified the misleading parts, and improved the manuscript accordingly. Therefore, I think the manuscript is suitable for publication in Solid Earth.

---

## Author Response (AR2)

**Anonymous reviewer:**

As a second-round reviewer, it is my honor to review the paper entitled "The Miocene subsidence pattern of the NW Zagros foreland basin reflects SE-ward propagating tear of the Neotethys slab". The evolution of a basin's accommodation space and the subsidence mechanism at convergent plate boundaries is a topic worthy of further investigation. In this paper, the authors focus on the basin located in the NW Zagros belt in the Kurdistan region of Iraq. By employing a synthesis of isopach maps, subsidence curves, regional Bouguer gravity anomaly, teleseismic tomographic data, and the magmatic record, the authors propose that the Zagros foreland basin subsided as a consequence of the combined loads exerted by the surface topography and the subducting slab during the early Miocene. Furthermore, they suggest that the basin was influenced by dynamic topography as a result of the Neotethys horizontal slab tear and northward flow of the Afar plume during the middle-late Miocene. In conclusion, the paper is well written and suitable for publication after minor revisions.

Dear second-round reviewer,
Thank you for the detailed reading and the helpful comments. We agree with all your points, and we have edited the manuscript accordingly.

Comments:
(1) I agree with the author that dynamic processes, such as slab tearing, played a role in the extra subsidence that occurred during the middle to late Miocene. But additional petrological evidence would be beneficial in order to support the hypothesis of slab tearing of the subducted Neo-Tethys lithosphere.

A paragraph addressing the petrological evidence has been added in the discussion section (5.2) as below:

Furthermore, when slab breakoff occurs, the geochemical properties of igneous rocks are anticipated to transition from primarily calc-alkaline (Ca-rich) to more alkaline (Na- and K-rich), accompanied by a shift in εNd values from negative to positive. This geochemical shift is attributed to the cessation of slab-derived fluids following slab breakoff. Similar patterns have been observed in eastern Turkey, where alkalinity has increased since the middle Miocene from north to south (Keskin, 2003; Şengör et al., 2008). In northwestern Iran, the geochemical compositions of the late Miocene-Quaternary Sablan, Sahand, and Saray volcanoes, which are located across the Arabia-Eurasia suture zone from northeast to southwest, respectively, in addition to slab subduction signatures, show medium to very high K content (ultrapotassic), positive εNd values (Sablan), and possibility of hot asthenospheric inflow (Saray) driving partial melting of an already metasomized mantle wedge (Moghadam et al., 2014; Ghalamghash et al., 2019; Chaharlang et al., 2023). Additionally, Miocene magmatism from northwest to southeast, from SE Turkey to NW Iran, shows εNd values shifting from positive to negative (Grosjean et al., 2022), suggesting that the magmas in SE Turkey originated from a primitive mantle melt (slab detached), whereas those in NW Iran resulted from a crustal contaminated melt (slab partially detached),

aligning with the hypothesized southeastward slab tearing proposed in this study (Figs. 7a,b, 10).

2) In section 5.2, the subheading is "Dynamic Subsidence and the Northward Flow of the Afar Plume." After reading this subheading, I was looking forward to seeing how the Afar plume affected subsidence. However, this section only emphasized in one sentence that "The early development of the northward flow of the Afar mantle material and its arrival at the Arabia-Eurasia suture zone by the middle Miocene possibly facilitated the slab tearing process". It would be beneficial if the authors can provide more details.

Thank you for your suggestion. We have added a new paragraph in the discussion section (5.2) to address this issue as below:

Taking into account that the collision between northern Arabia and Eurasia occurred during the early Oligocene, slab mechanical weakening and necking were already underway before the arrival of the Afar asthenospheric mantle material during the middle Miocene. Three-dimensional analogue models simulating the Arabia-Eurasia collision suggest that slab tearing began in northwestern Arabia due to the lateral transition of the Arabian continental crust into the oceanic crust toward the Mediterranean region. This transition led to the subduction of the oceanic plate at the Eurasian margin while the continental plate collided, due to buoyancy differences (Faccenna et al., 2006). The oblique convergence between Arabia and Eurasia (McQuarrie et al., 2003; Navabpour et al., 2008) may have further promoted slab tearing, as this obliquity caused an earlier continent-continent collision compared to adjacent regions where oceanic subduction continued, as demonstrated by numerical models (Boonma et al., 2023). During the middle Miocene, the arrival of the hot asthenospheric mantle likely accelerated slab tearing by thermally weakening the slab necking zone, reducing its viscosity and strength (Keskin, 2007; Menant et al., 2016; Boutelier and Cruden, 2017). By the late Miocene, as the southeastward horizontal tearing of the slab progressed, the northward flow of the Afar plume asthenospheric mantle material advanced into eastern Anatolia. This process triggered a dynamic uplift along western Arabia and the TIP, accompanied by dynamic subsidence on the east side (e.g., Daradich et al., 2003; Craig et al., 2011). Additionally, as the Neotethys oceanic plate remained attached to the Arabian continental plate on its eastern side, several numerical models indicate that mantle downward flow along the subducting slab also contributes to the dynamic subsidence (e.g., Bottrill et al., 2012; Duretz and Gerya, 2013; Balázs et al., 2022).

(3) Line 119 N—north, NE—northeast
(4) Line 184 and 186, 188,189 Formations—formations
(5) The font size of the latitude and longitude in figures 3 and 6 is too small.
Points 3-5 were edited accordingly.

(6) What does the yellow color in Figure 7A represent?
Neogene basin isopach maps as written in the caption, but for clarity, a legend is now provided.

(7) Maybe I don't have a good sense of three-dimensionality. Is the direction of N in Figure 10 correct? In addition, I suggest that the authors divide the subducted slab into a lower subducted Neotethys oceanic slab and an upper Arabian continental slab.

Thank you. The N arrow has been modified. Additionally, we colored the continental and oceanic plates differently and edited labels.

**Yang Chu, Editor**
I have received two reviews, one from previous reviewer, and one new reviewer. I have checked the comments carefully and find that only minor revision is required before acceptance. I agree with the second reviewer that some part still needs clarification. After that, I think this manuscript can be accepted. This work would bring new insights into the geodynamic evolution of the subduction of the NeoTethys. Here I also give my suggestions.

Dear Yang Chu, Editor,
Thank you for your careful reading and comments. We have edited the manuscript accordingly.

Line 13: 3-4. Also change it elsewhere.
Line 34: I am not familiar with this mantle orogen. To my knowledge, this kind is generated by basal shear of asthenospheric mantle flow, and mantle here can lithospheric or asthenosphere. mantle.
Line 37-39: foreland basin formation is also part of orogeny.
Line 61: why is thermal emphasized here?
Line 62: slow vs fast? Do you mean tomography? Or low and high velocity bodies?
Line 77-78: this sentence has no meaning here. Delete it. (This was not clear which sentence)
Line 110: afterwards
Thank you for pointing out the issues above; they are fixed accordingly.

Line 269, figure 7: put numbers on the profiles.
The profiles in Fig. 7 are labeled as I, II, III

I am also curious if magmatism by the slab tearing can be marked on any figure. This will demonstrate directly the tearing.
figure10: there is a long section on the Afar influence, while it is not shown on the final model.

Thank you for your comment. We have added a new paragraph in the discussion section (5.2) to address magmatism in more detail (see the response to the anonymous reviewer). We also updated Fig. 10 to reflect the content better; however, we prefer to keep it simple and more conceptual.